# High-fidelity spatial mode transmission through a 1-km-long multimode fiber via vectorial time reversal

Yiyu Zhou[1✉], Boris Braverman[2], Alexander Fyffe[3], Runzhou Zhang[4], Jiapeng Zhao[1], Alan E. Willner[4], Zhimin Shi[3] & Robert W. Boyd[1,2]

The large number of spatial modes supported by standard multimode fibers is a promising platform for boosting the channel capacity of quantum and classical communications by orders of magnitude. However, the practical use of long multimode fibers is severely hampered by modal crosstalk and polarization mixing. To overcome these challenges, we develop and experimentally demonstrate a vectorial time reversal technique, which is accomplished by digitally pre-shaping the wavefront and polarization of the forward-propagating signal beam to be the phase conjugate of an auxiliary, backward-propagating probe beam. Here, we report an average modal fidelity above 80% for 210 Laguerre-Gauss and Hermite-Gauss modes by using vectorial time reversal over an unstabilized 1-km-long fiber. We also propose a practical and scalable spatial-mode-multiplexed quantum communication protocol over long multimode fibers to illustrate potential applications that can be enabled by our technique.

[1] The Institute of Optics, University of Rochester, Rochester, NY, USA. [2] Department of Physics, University of Ottawa, Ottawa, ON, Canada. [3] Department of Physics, University of South Florida, Tampa, FL, USA. [4] Department of Electrical and Computer Engineering, University of Southern California, Los Angeles, CA, USA. ✉email: yzhou62@ur.rochester.edu

Quantum cryptography[1,2] is a maturing technology that can guarantee the security of communication based on the fundamental laws of physics. The secure key rate of quantum key distribution (QKD) systems is many orders of magnitude lower than the data transfer rate of classical communication systems, inhibiting the widespread adoption of QKD in practical scenarios. Numerous methods have been proposed to increase the secure key rate in QKD, such as the development of new protocols[3], use of high-performance detectors[4], and wavelength-division multiplexing[5]. The spatial degree of freedom in a multimode fiber (MMF) has long been recognized as an additional resource to further increase the communication rate by either mode-division multiplexing[6–13] or high-dimensional encoding[14–18]. It is compatible with other multiplexing methods such as wavelength-division multiplexing and can be also used to enhance quantum teleportation[19] and entanglement distribution[20,21]. However, the inevitable mode crosstalk in standard MMFs is a persistent obstacle to practical applications of spatial modes for QKD. Tremendous efforts have been devoted to attempts to mitigate the effects of spatial-mode crosstalk during the past decades. Transfer matrix inversion is a standard method that has been successfully used to transmit spatial modes through MMFs[22–28]. However, standard MMFs can support between tens and hundreds of modes depending on the wavelength, and thus the number of complex-valued elements in the transfer matrix is typically between $10^3$ and $10^5$. As a consequence, all transfer matrix inversion experiments reported in the literature have used a short MMF ($\approx 1$ m)[22–28] because the fiber has to be carefully stabilized during the slow characterization process (see Supplementary Note 1 for a summary of the fiber length used in previously reported experiments). When applying this method to a long fiber, it is foreseeable that instability will severely impede long-distance communication outside the laboratory. By contrast, mode-group excitation[29–31] has been applied to long fibers due to the relatively low inter-modal-group crosstalk. However, for a fiber supporting $N$ spatial modes, only approximately $\sqrt{N}$ mode groups are supported. Thus the number of usable mode groups is intrinsically limited in this method (see Supplementary Note 1). Multiple-input-multiple-output (MIMO) algorithm is another standard method for classical crosstalk mitigation[32]. However, it requires a high signal-to-noise ratio for digital signal processing and thus is unsuitable for quantum applications. Hence, none of these existing methods can be used to support a large number (>100) of modes for high-dimensional or spatial-mode-multiplexed QKD over long, unstabilized, standard MMFs outside the laboratory.

Optical time reversal, which is also referred to as phase conjugation[33], is an alternative method for modal crosstalk suppression. The concept of time reversal is illustrated in Fig. 1a. The wavefront of an optical beam transmitted by Bob is distorted by an aberrating medium as shown in the left panel. Reflecting the beam by an ordinary mirror at Alice's side is not helpful as the wavefront becomes distorted even more severely (see the middle panel). In contrast, a time-reversing mirror flips the sign of phase of the reflected beam, and consequently wavefront distortion can be exactly corrected after propagating through the same aberrating medium[33] as illustrated in the right panel. Although we use a simple plane wave to illustrate the concept, it should be noted that time reversal is applicable to an arbitrary spatial mode. Optical time reversal has been investigated for biological tissues[34–44] and MMFs[41–46]. The textbook description of time reversal[33] assumes a scalar incident field $\mathbf{E}_{\text{scalar}}(\mathbf{r}) = \hat{\epsilon} A(\mathbf{r}) e^{i\mathbf{k}\cdot\mathbf{r}}$, where $\hat{\epsilon}$ is the polarization unit vector, $A(\mathbf{r})$ is the complex field amplitude, $\mathbf{r}$ is the position vector, and $\mathbf{k}$ is the wavevector of the incident field. It can be seen that the polarization $\hat{\epsilon}$ and the complex field amplitude $A(\mathbf{r})$ are separable from each other. The scalar time reversal of the incident field can be expressed as[33]

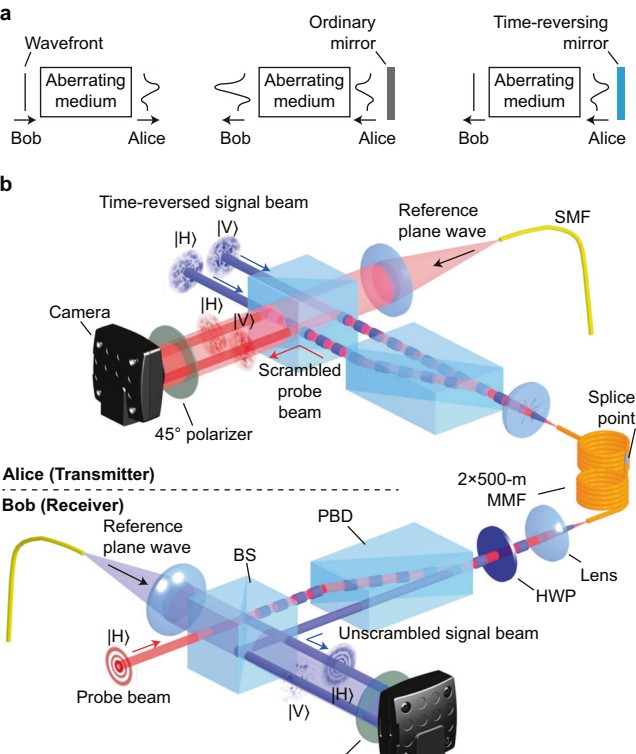

**Fig. 1 Illustration of experiment. a** The concept of time reversal. **b** Schematic of the experiment to send high-fidelity spatial modes from Alice to Bob. Bob first transmits a probe beam to Alice (denoted by red beams). Alice performs vectorial off-axis holography on her received probe beam and generates the corresponding time-reversed signal beam (denoted by blue beams). |H⟩ and |V⟩ stand for the horizontal and vertical polarization state, respectively. HWP: half-wave plate. PBD: polarizing beam displacer. SMF: single-mode fiber. BS: beamsplitter. See Supplementary Note 2 for experimental details.

$$\mathbf{E}_{\text{scalar}}^*(\mathbf{r}) = \hat{\epsilon}^* A^*(\mathbf{r}) e^{-i\mathbf{k}\cdot\mathbf{r}}, \quad (1)$$

where * in the superscript denotes the complex conjugate. Here we generalize the scalar time reversal and propose the vectorial time reversal. Assume a vectorial incident field given by $\mathbf{E}_{\text{vector}}(\mathbf{r}) = \hat{x} A_1(\mathbf{r}) e^{i\mathbf{k}_1\cdot\mathbf{r}} + \hat{y} A_2(\mathbf{r}) e^{i\mathbf{k}_2\cdot\mathbf{r}}$, where $A_1(\mathbf{r})$ and $A_2(\mathbf{r})$ ($\mathbf{k}_1$ and $\mathbf{k}_2$) denote the respective complex field amplitude (wavevector) of the horizontally and vertically polarized field, and $\hat{x}$ ($\hat{y}$) denotes the horizontal (vertical) unit vector. Its vectorial time reversal can be written as

$$\mathbf{E}_{\text{vector}}^*(\mathbf{r}) = \hat{x} A_1^*(\mathbf{r}) e^{-i\mathbf{k}_1\cdot\mathbf{r}} + \hat{y} A_2^*(\mathbf{r}) e^{-i\mathbf{k}_2\cdot\mathbf{r}}. \quad (2)$$

Here, the incident vectorial field is described by a non-separable state, which is a more general form than a separable state. All experimental demonstrations in MMFs[41–46] to date have been solely based upon scalar time reversal. However, the scalar time reversal can only succeed in a short MMF ($\approx 1$ m)[41–45] or a few-mode fiber[46], because the optical field scrambled by a long MMF has to be described by a vectorial field due to the spatially-varying birefringence in a MMF.

In this work, we experimentally show how vectorial time reversal can be used to support 210 modes over a 1-km-long MMF. The fiber length used in our experiment is nearly three orders of magnitude greater than those used in many previously reported experiments demonstrating high-fidelity spatial-mode transmission through MMFs (see Supplementary Note 1). To illustrate the potential of our method, a spatial-mode-multiplexed

QKD protocol is proposed that is suitable for realization in real-world, unstable links with a practical, scalable implementation.

## Results

**Schematic of vectorial time reversal.** Figure 1b presents the conceptual schematic of our experiment. Here we experimentally demonstrate that digital vectorial time reversal can be successfully applied to transmit 210 high-fidelity spatial modes (up to mode group 13) through a 1-km-long, standard, graded-index MMF with the number of used modes limited by the active area size of our SLM. We choose the commonly used Laguerre–Gauss and Hermite–Gauss modes for demonstration because they are the eigenmodes of a graded-index MMF[47], which exhibit better robustness and minimized loss during propagation[10] compared to other basis sets and thus enable time reversal to a full extent. A 780 nm laser is used as the light source in the experiment. Bob prepares a spatial mode of interest (i.e., a probe beam) and transmits it to Alice through a 1-km-long, standard, graded-index MMF (Clearcurve OM3, Corning). The fiber is comprised of two 500-m-long bare fibers that are spliced together and are free of any specialized thermal or mechanical isolation. The fiber has a core diameter of 50 $\mu$m and NA = 0.2, therefore supporting ≈400 modes per polarization at 780 nm. The polarization of the probe beam transmitted by Bob can be adjusted by a half-wave plate. After transmission through the fiber, the scrambled probe beam received by Alice has a distorted spatial and polarization profile. Alice then performs vectorial off-axis holography to measure the spatial and polarization profile of the scrambled probe beam. First, Alice uses a polarizing beam displacer to coherently separate the horizontally and vertically polarized components of the scrambled probe beam into two beams that propagate along the same direction but are transversely displaced with respect to each other. These two beams are then combined with a coherent, 45° polarized reference plane wave at a beamsplitter, and the resultant interference pattern is recorded by a camera. Through off-axis holography[48], the amplitude, phase, and polarization of the scrambled probe beam can be simultaneously determined via a single-shot measurement[49] (see Supplementary Note 2). Alice then uses a single spatial light modulator (SLM) to generate the back-propagating signal beams, which are the phase conjugate of the displaced, scrambled probe beams. The two back-propagating time-reversed signal beams are combined coherently by the same polarizing beam displacer. After passing through the same MMF, the signal beam is unscrambled and becomes the mode originally transmitted by Bob with a reversed wavefront. Vectorial off-axis holography is then performed by Bob to quantitatively characterize the spatial and polarization profile of the unscrambled signal beam. Additional experimental details are provided in Supplementary Note 2.

**Characterization of modal fidelity.** Figure 2a, b shows two examples of experimentally measured scrambled probe beams received by Alice and the unscrambled signal beams received by Bob for horizontally polarized LG(3,2) and HG(4,4) modes, where the mode indices of Laguerre–Gauss mode are denoted by LG($p,\ell$) and that of Hermite–Gauss mode are denoted by HG($m,n$). It can be seen that the scrambled probe beams are vectorial fields and cannot be described by a separable state, because field profiles of horizontal and vertical polarization are very different. For each unscrambled signal beam received by Bob, we digitally project the mode to an orthonormal spatial-mode basis set to calculate the crosstalk matrix. We measure the crosstalk matrix for 105 Laguerre–Gauss modes with $2p + |\ell| \leq 13$ in both horizontal and vertical polarization basis sets, resulting in a $210 \times 210$ crosstalk matrix. The same measurement is also performed for

Hermite–Gauss modes with $m + n \leq 13$. The unnormalized modal fidelity for individual spatial modes (i.e., the diagonal elements of crosstalk matrix) is shown in Fig. 2c with an average of 85.6% for Laguerre–Gauss modes and Fig. 2d with an average of 82.6% for Hermite–Gauss modes. Here the crosstalk matrix element is calculated as $M_{k',k} = |\langle \phi_{k'}^{\text{ideal}} | \phi_k^{\text{exp}} \rangle|^2$ and the unnormalized modal fidelity is $F_k = M_{k,k}$, where $|\phi_{k'}^{\text{ideal}}\rangle$ is the ideal spatial mode with a mode index of $k'$, $|\phi_k^{\text{exp}}\rangle$ is the experimentally measured spatial mode with a mode index of $k$, $\langle \phi_{k'}^{\text{ideal}} | \phi_{k'}^{\text{ideal}} \rangle = 1$ and $\langle \phi_k^{\text{exp}} | \phi_k^{\text{exp}} \rangle = 1$. The normalized modal fidelity within the 210-mode subspace has an average of 91.5% for Laguerre–Gauss modes and 89.3% for Hermite–Gauss modes, where the normalized modal fidelity is defined as $F_k^{\text{norm}} = M_{k,k} / \sum_{k'=0}^{209} M_{k',k}$. The full $210 \times 210$ crosstalk matrices for both Laguerre–Gauss and Hermite–Gauss modes are provided in Supplementary Note 4. Although the performance is characterized using a classical light source and a classical detector, our method is readily applicable to QKD by attenuating the light intensity to a single-photon level and by using single-photon detectors[50]. The additional noise of single-photon detectors is outside the scope of this work. Here we attribute the imperfect modal fidelity mainly to the imperfect mode generation by the SLM. To test this hypothesis, we experimentally characterize the fidelity of the probe beam generated by Bob and that of the signal beam generated by Alice. The product of these two fidelities is referred to as experimental generation fidelity, which is presented as solid lines in Fig. 2c, d for individual spatial modes (see Supplementary Note 5 for details). It can be seen that the fidelity of unscrambled signal beams received by Bob matches well with the experimental generation fidelity. Hence, the fidelity of unscrambled signal beams is limited by our experimental apparatus (e.g., the SLM) rather than the fiber length, and we believe that our method can be applicable to even longer fibers. We also note that such high modal fidelity is exclusively enabled by vectorial time reversal, while scalar time reversal can only achieve an average unnormalized modal fidelity of 41.2% for Laguerre–Gauss modes and 39.7% for Hermite–Gauss modes (see Supplementary Note 6). Nonetheless, there exists a deviation between the unscrambled modal fidelity and the experimental generation fidelity for high-order modes, which we attribute to the fact that high-order modes are susceptible to mode-dependent loss induced by fiber bending and splicing. Further studies are needed to identify the exact reason in order to further expand the transmission distance for high-order modes. To evaluate the performance of our system for polarization-based QKD, we measure the $4 \times 4$ polarization crosstalk matrix for each mode within the corresponding spatial-mode subspace. The resultant normalized polarization crosstalk matrices for LG(0,2), LG(1,2), LG(2,2), and LG(3,2) are shown in Fig. 3a. The average polarization crosstalk is 0.04% for Laguerre–Gauss modes and 0.05% for Hermite–Gauss modes (see Supplementary Note 7), which suggests that both the spatial mode and polarization scrambling can be well suppressed through vectorial time reversal. These high-fidelity results directly indicate that the polarization-based QKD protocol can be performed through MMFs, and the secure key rate can be significantly boosted by mode-division multiplexing. Furthermore, because these high-fidelity results are obtained in a spliced fiber, we expect that the vectorial time reversal would also be realized in a much longer fiber. We also calculate the crosstalk matrix for the scrambled probe beams received by Alice in the absence of vectorial time reversal, and the average unnormalized modal fidelity in this case is ≈ 1% for both Laguerre–Gauss modes and Hermite–Gaussian modes (see Supplementary Note 4), which shows the strong mode scrambling in fiber and by contrast highlights the effectiveness of our method.

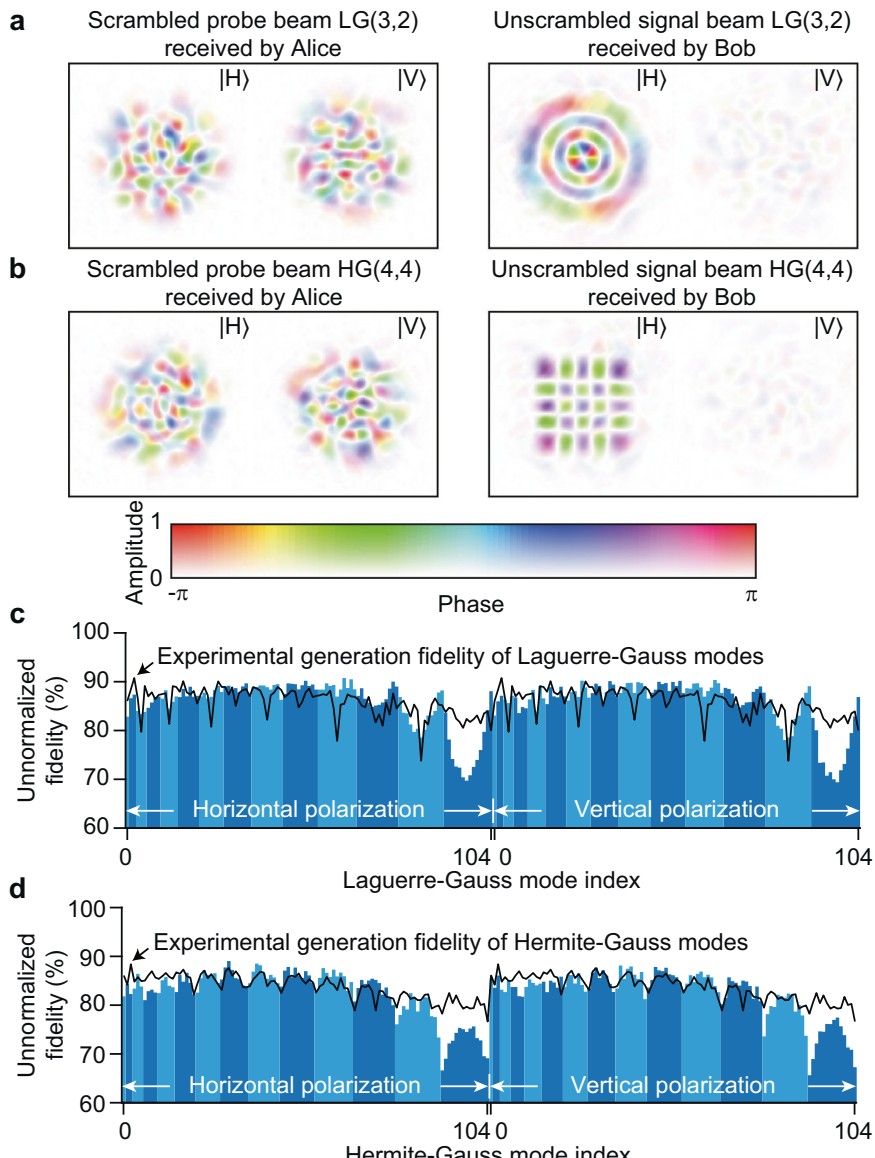

**Fig. 2 Modal fidelity measurement. a**, **b** The measured amplitude, phase, and polarization of the scrambled probe beams and unscrambled signal beams for horizontally polarized LG(3,2) and HG(4,4) mode respectively. **c**, **d** The unnormalized modal fidelity for unscrambled Laguerre–Gauss modes and Hermite–Gauss modes. A single index is used to denote the two-mode indices for simplicity (see Supplementary Note 3), and the light and dark blue bands denote the odd and even mode group number respectively.

**Response time of the vectorial time-reversal system**. To overcome environmental instability, which is an inevitable concern for long fibers in a real-world environment, vectorial off-axis holography needs to be performed repeatedly in real time, and the phase pattern on Alice's SLM should be updated accordingly to compensate for instability. In the following, we evaluate the response time of our vectorial time-reversal system. To perform off-axis holography, we need to retrieve a single-shot image from the camera (which takes 2.6 ms) and execute fast Fourier transforms and interpolations as digital signal processing (which takes 18 ms on a desktop computer, see Supplementary Note 2). It should be noted that the data processing time can be significantly reduced by using a dedicated digital signal processor or even eliminated by careful experimental design and alignment as discussed in Supplementary Note 2. The response time of our system is therefore only constrained by the refresh rate of SLM, which is 4 Hz in our experiment. However, this constraint can be readily removed by using a commercially available fast digital

micromirror device (above 10 kHz refresh rate[51]) or a high-speed SLM (sub-kHz refresh rate[52]). In addition, we emphasize that the data transfer rate using each spatial mode is not limited by the refresh rate of SLM or the response time of the time-reversal system but is determined by the modulation rate of the modulator used by the encoder[53], and the time-reversal system response time only needs to be faster than the environmental fluctuation rate in order to overcome instability. In this proof-of-principle experiment, we implement digital vectorial time reversal for one spatial mode at a time, but we emphasize that our method can be readily used to enable mode-division multiplexing[46]: by separately pre-shaping individual wavefronts of multiple high-speed-modulated signal beams, high-fidelity spatial modes can be recovered at the receiver, and thus the data streams can be demultiplexed with low crosstalk. To test the operation stability of our system, we also measure the unnormalized modal fidelity as a function of time while the SLM is actively updated every ≈30 seconds for each mode, which is depicted by the solid

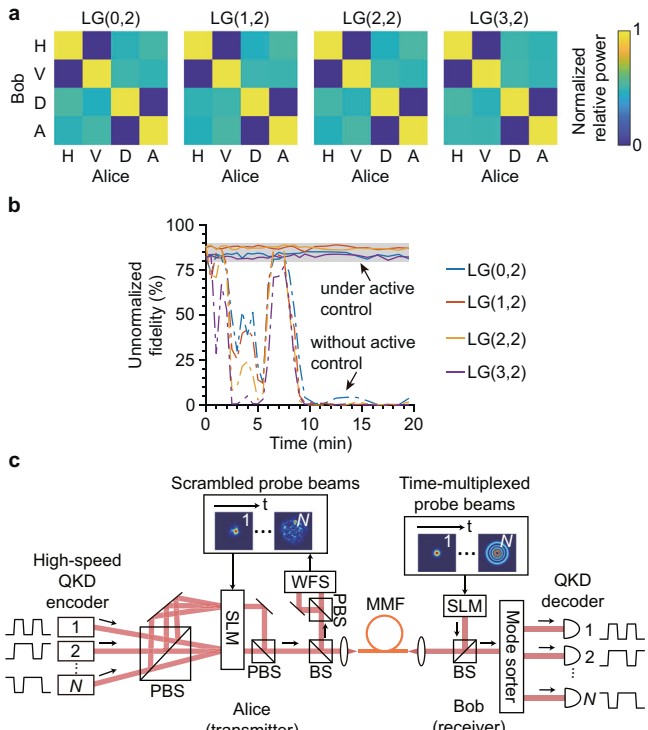

**Fig. 3 System performance evaluation. a** The normalized crosstalk matrix in the polarization subspace for horizontal (H), vertical (V), diagonal (D), and anti-diagonal (A) polarizations for LG(0,2), LG(1,2), LG(2,2), and LG(3,2) modes. **b** Stability test for vectorial time reversal. The unnormalized modal fidelity is measured as a function of time. The shaded area corresponds to modal fidelity between 80% and 90%. The solid lines represent the results when the SLM is under active control and the dashed lines represent the results without active control. **c** Proposed spatial-mode-multiplexed QKD protocol. A densely encoded computer-generated hologram imprinted on a single SLM can be used to simultaneously generate and multiplex a large number of spatial modes. BS: beamsplitter.

lines in Fig. 3b; here the dashed lines represent the modal fidelity in the absence of active control of the SLM. The autocorrelation $R(\Delta t) = \langle|\langle\phi(t)|\phi(t+\Delta t)\rangle|^2\rangle$ is calculated according to these data, where $|\phi(t)\rangle$ is the time-reversed mode at time $t$, $\langle\cdot\rangle$ is the time average, and $|\phi(t)\rangle$ is normalized such that $\langle\phi(t)|\phi(t)\rangle = 1$. The time for $R(\Delta t)$ to drop to $1/e$ is approximately 120 s for LG (0,2) and 100 s for LG(3,2). These results clearly show that our system is able to overcome environmental instability even though the unprotected 1-km-long bare fiber is placed on an optical table that has not been floated and is free of any thermal or mechanical isolation. The image of the unprotected fiber spool is shown in Supplementary Note 2. We believe that by using a fast SLM, real-time crosstalk suppression can be achieved even in a harsh environment through a much longer fiber.

**Proposed quantum communication protocol.** Figure 3c presents a practical, scalable QKD protocol with mode-division multiplexing. Each spatial mode can be used as an independent channel, with time-bin encoding[54–57] or continuous-variable encoding[58–60] used to guarantee communication security within each channel. This spatial-mode-multiplexed setup can also be used to implement high-dimensional encoding by transmitting one spatial mode at a time. The Laguerre–Gauss and Hermite–Gauss modes can be employed as mutually partially unbiased bases for high-dimensional encoding to guarantee security[18]. In particular, it has been previously demonstrated that

a single SLM can be used to simultaneously generate and multiplex 105 spatial modes by using a densely encoded computer-generated hologram[61]. To implement the $N$-mode-multiplexed QKD protocol, Bob first sequentially transmits $N$ probe beams of interest to Alice through the fiber via spatial-mode switching, which can be realized by an acousto-optic modulator with a mode switching rate up to 500 kHz[62]. Alice measures the scrambled probe beams using a wavefront sensor (WFS), computes the densely encoded hologram, and then imprints the hologram onto her SLM. Although the densely encoded hologram has a low efficiency, this will not reduce the secure key rate of the attenuated-coherent-state-based QKD because strong loss is inherently necessary to attenuate a classical laser to the single-photon level. The WFS can be realized by either the off-axis holography presented in this work or alternative methods such as the commercial Shack–Hartmann WFS and the vectorial complex field direct measurement[49]. Alice then prepares $N$ attenuated signal beams with high-speed time-bin encoding or continuous-variable encoding and illuminates the SLM with these beams incident at different angles. These $N$ beams can be obtained by using a single laser with a 1-to-$N$ fiber beamsplitter. The loss induced by the beamsplitter at the transmitter's side is not a concern for attenuated-coherent-state-based QKD protocols. The horizontal and the vertical polarization components are split by a polarizing beamsplitter (PBS) and are incident at two separation locations on the SLM, which allows for the generation of vectorial time reversal. Alice's SLM converts each of the $N$ incident beams into the phase conjugate of its corresponding scrambled probe beam, in addition to multiplexing all the modes to propagate in the same direction. The horizontal and vertical polarization components are recombined by another PBS and finally transmitted to Bob through the MMF. As a consequence, all channels can have a high-fidelity spatial profile at Bob's side, and Bob can use the well-developed Laguerre–Gauss or Hermite–Gauss mode sorter[63–65] to demultiplex the signal beams.

## Discussion

The loss induced by the beamsplitter at the receiver's side can be reduced by using a high-transmission beamsplitter (such as a T:R = 95:5 beamsplitter). We note that the reduced secure key rate caused by this small amount of additional loss can be well compromised by the capacity improvement of mode-division multiplexing. To overcome environmental instability, Bob needs to periodically send probe beams of interest to Alice, who updates the phase pattern on the SLM accordingly. The polarization degree of freedom can also be included to further increase the channel capacity. It should be noted that the signal transfer speed is determined by the QKD encoder, not the SLM refresh rate. We emphasize that an analogous protocol, to the best of our knowledge, cannot be realized in a straightforward manner by any alternative methods for the following reasons. Since a complete knowledge of the complex-valued transfer matrix is not needed, our method can be applied to unstabilized, long MMFs outside the laboratory, which is not possible by slow, conventional transfer matrix inversion. MIMO is not applicable to QKD because it requires a large number of photons for digital signal processing. Mode-group excitation only allows for a small number ($\approx$10) of mode groups and is thus unable to fully utilize the channel capacity of the link. Thus, vectorial time reversal offers a unique and practical approach towards spatial-mode-multiplexed quantum communication over realistic, unstable links.

In summary, we have demonstrated that, through the use of vectorial time reversal, we can establish a high-fidelity, 1-km-long communication link that supports 210 spatial modes of a standard MMF. Both spatial-mode crosstalk and polarization scrambling in

MMF can be well suppressed, which demonstrates the possibility of boosting the communication rate of both classical communication and QKD by either mode-division multiplexing or high-dimensional encoding. In particular, we propose a spatial-mode-multiplexed QKD protocol and show how our method can be used to boost the channel capacity in a straightforward manner. While specialty fibers such as multi-core fibers[66] can also be used to realize space-multiplexed QKD, standard MMFs are cheaper and already widely deployed in existing commercial fiber communication systems, and thus the capacity of these systems can be readily improved by two orders of magnitude using our method without replacing the fibers. Our results also confirm the time-reversal symmetry of beam propagation over a 1-km-long MMF. Given the scalability of the experimental implementation and high fidelity of the data, our technique can be useful to not only mode-division multiplexing but also to other applications such as fiber endoscopy[67], lensless microscopy[68], and high-dimensional entanglement distribution[20,21].

## Data availability

The data that support the findings of this study are available from the corresponding author upon reasonable request.

## Code availability

All relevant computer codes supporting this study are available from the corresponding author upon reasonable request.

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

## Acknowledgements

This work is supported by the U.S. Office of Naval Research (N00014-17-1-2443, N00014-20-1-2558, N00014-16-1-2813). B.B. acknowledges the support of the Banting Postdoctoral Fellowship. R.W.B. acknowledges funding from the Natural Sciences and Engineering Research Council of Canada, the Canada Research Chairs program, and the Canada First Research Excellence Fund. A.E.W. acknowledges the support of the Vannevar Bush Faculty Fellowship sponsored by the Basic Research Office of the Assistant Secretary of Defense for Research and Engineering. R.Z. acknowledges the support of the Qualcomm Innovation Fellowship.

## Author contributions

Y.Z. conceived and performed the experiment with assistance from B.B., Z.S., R.Z., J.Z. and R.W.B. Y.Z., B.B., A.F., R.Z., J.Z., A.E.W., Z.S. and R.W.B. contributed to the discussion of the results and the writing of the manuscript. A.E.W., Z.S. and R.W.B. supervised the project.

## Competing interests

The authors declare no competing interests.
