## [Peer Review File · Nature Communications]

Reviewers' Comments:

Reviewer #1:

Remarks to the Author:

The authors have demonstrated the use of vectorial time reversal to transmit several spatial photonic modes over a 1 km multi-mode fiber. The employed technique requires the use of an additional laser transmitting the specific mode in question from the receiver (Bob) to the transmitter (Alice), and based on the detected spatial profile, a pre-compensation transformation is employed to the transmitted signal from Alice to Bob in order to undo the spatial distortions and mode coupling caused by the fiber. The technique is successful enough generating decent fidelities across most modes, and low enough crosstalk to enable parallelisation of QKD in the same fiber. I find the capability of being able to transmit so many modes over 1 km very impressive. The paper is generally well written and the figures are clear.

On the other hand, the obtained results as well as the added required hardware do not directly support two major claims in the abstract in my opinion, namely: "allowing for a channel capacity of up to 13.8 bits per sifted photon for high-dimensional quantum communication" and "We also propose a practical and scalable mode- multiplexed QKD protocol that cannot be achieved by alternative methods." I am therefore unable to give a final recommendation unless the authors are able to clarify my concerns below:

- In itself, I do indeed find impressive the capability to transmit so many modes across such a long fiber, and I think this is quite an advance. However I am not convinced that the two main applications mentioned by the authors can be directly carried out. Since these two applications work very differently, and for the sake of clarity, I will address each separately.

(a) In the case of spatial high-dimensional quantum communication, and taking BB84 as an example, it is required that Alice chooses randomly between one of the two bases (HG or LG) and between any of the 210 modes in that basis. The issue is that then the time reversal system cannot transmit that specific mode from Bob to Alice, since it is not known a priori which mode it would be. And of course, if the mode to be sent was known by Bob, then there is no need to do anything since there is no information transfer. For me the problem lies in the fact that a specific mode needs to be transmitted first from Bob to Alice, and since Alice needs to send a random attenuated spatial mode back, the time reversal compensation would not work. Therefore it is unclear to me how HD spatial QKD could work with this setup, unless if there is a single compensation that can be simultaneously applied for all modes to be transmitted. Please see my comment in the conclusion with a possible solution.

(b) The case for multiplexed-QKD is a bit more clear, but I still have some concerns. This idea in itself is not novel, and this was explored already in Bacco et al, *Comm. Phys.* 2, 140 (2019). There they used each spatial mode of a multi-core fiber to transmit a separate QKD stream, and then boosted the overall rate by a factor of 37. Therefore, I don't quite agree with the claim that "cannot be achieved by alternative methods", as this has already been done. Granted, the use of multi-mode fiber could boost the key even further by offering more spatial channels, but it wouldn't be the first time it was done.

- One minor suggestion I have in the supplemental material is that the authors put the figures in line with the text where they are cited, as it makes the reading easier.

In conclusion:

I think the authors have a very nice result, but it is not very clear how it can be applied for other applications. The main issue for me is the requirement that each mode needs to be specifically transmitted from Bob to Alice, and that then a new beam has to be generated for that particular

mode.

In the case of (a), the authors need to show a way to make the QC protocol working. One possibility could be to apply the compensation technique from (b) such that N different lasers would be present at Alice, and depending on which state to be sent, a different laser (corresponding to each mode) would be fired (like free-space BB84 systems where four laser diodes are used (one for each polarization state)). Maybe even using a VCSEL array perhaps, to avoid having hundreds of lasers. Or a single laser with an acousto-optic modulator? Or something else more elegant?

In the case of (b) I agree it looks feasible, but the impact is much more limited than 2 orders of magnitude, since it was already done gaining more than 1 order of magnitude compared to standard QKD systems (D. Bacco et al).

Reviewer #2:

Remarks to the Author:

Dear Editor,

The paper reports an experimental study of multi-modal propagation in 1-km fiber in a context of quantum communication. The Authors describe the method named "vectorial time reversal", which assumes that the mode transformation is first measured using off-axis holography and then the wavefront of the field is modified using spatial light modulator to reverse the effect of mode mixing. The method is demonstrated experimentally using laser light. It is also analyzed for the application in quantum key distribution. The Authors claim the capacity of the channel of the order of 13 bits per sifted photon.

Here is a list of my comments:

- 1) The Authors present a classical method which can be applied to enhance quantum communication protocols including QKD. In my opinion this is nice piece of engineering, but I do not see new physics.
- 2) In practice, photon loss is crucial for QKD applications. The method is based on adaptive optics, in particular on SLM, which are very lossy. One can argue that deformable mirror can be used, with much higher reflection and also refresh rate, but with the limited resolution. This is an engineering problem that should be addressed properly if this method is to be proved useful in practice.
- 3) On page 4, just at the beginning it is said "we believe that the fidelity of time-reversed modes can be further increased by using a well-calibrated SLM". Why the proper calibration has not been done?
- 4) "In this proof-of-principle experiment, we implement digital vectorial time reversal for one spatial mode at a time, but we emphasize that our method can be readily used to enable mode-division multiplexing". It would be interesting, from the perspective of practical application of the method, what is the expected performance when available adaptive optics is used (refresh rate impact mainly).
- 5) There is multiple trade-offs, which are not discussed, including: the photon loss decreases the efficiency of QKD but the channel capacity is increased by using higher spatial modes.
- 6) In the paper [Optics Express, 2019, 27, 37214] the authors present a passive method of reducing an effect of multi-modal propagation. It is related to free space propagation, but the underlying physics is related.

Minor comments:

- 1) Page 1, column 2 "However, it require a high signal-to-noise ratio..." should be "requires"
- 2) page 2, column 1 "Figure 1(b) presents the the conceptual schematic.. " one "the" should be dropped.

Summarizing, the paper and supplementary material are very clearly written. The presented method does not offer any significant improvement for quantum communication protocols (eg for QKD). Some further analysis is required. I do not recommend this manuscript to publish in Nature Communication. I suggest much more specialized journal.

Best regards,
A Referee

Reviewer #3:

Remarks to the Author:

The paper "High-fidelity spatial mode transmission through a 1-km-long multimode fiber via vectorial time reversal", presents an experimental demonstration of a rudimental scheme for quantum key distribution (QKD).

The authors would like to exploit the space encoding technique for transmitting high-dimensional quantum states over 1 km of multimode fiber combined with space-division multiplexing.

Although the work looks interesting for the entire quantum community and the experiment is documented with rigour, this paper is not demonstrating any advantage compared to standard QKD scheme. The secret key is not measured, neither extrapolates since all the measurements are done in the classical domain and the scalability of the entire scheme is not feasible.

My main concern is that the authors claim to do QKD without using single photon detectors at all. QKD does not only work by attenuating a laser and doing classical measurements.

One of the main difficulties of high-dimensional quantum key distribution is to prepare quantum states in real-time. All the claims about high-rate and good fidelity are not sufficient since when you are preparing your quantum states in real-time you will have different errors and different performances. Moreover, the authors proposed a scheme for multiplexing which in principles should work, but then they don't have a single QKD system to be tested on their experiment. As such I recommend the authors to carefully consider the journals in which they would like to publish their work.

Here I attached some more detailed comments:

1. The quantum states which are used are not clearly reported in the manuscript.
2. Decoy state technique is not included at all in your analysis. Why not?
3. A stability measurement over time is missing in the paper. How is drifting the system?
4. In the quantum systems the losses are very important parameters? How many losses do you have at your receiver?
5. Since the authors are claiming to use this setup for QKD, how would you modulate these optical modes? With a spatial-light-modulator at a very low-speed? Do you have other solutions?

Response Letter

Report of reviewer #1 (in blue) followed by a detailed response to each point (in black)

Reviewer #1 — comment #1: The authors have demonstrated the use of vectorial time reversal to transmit several spatial photonic modes over a 1 km multi-mode fiber. The employed technique requires the use of an additional laser transmitting the specific mode in question from the receiver (Bob) to the transmitter (Alice), and based on the detected spatial profile, a pre-compensation transformation is employed to the transmitted signal from Alice to Bob in order to undo the spatial distortions and mode coupling caused by the fiber. The technique is successful enough generating decent fidelities across most modes, and low enough crosstalk to enable parallelisation of QKD in the same fiber. I find the capability of being able to transmit so many modes over 1 km very impressive. The paper is generally well written and the figures are clear.

On the other hand, the obtained results as well as the added required hardware do not directly support two major claims in the abstract in my opinion, namely: "allowing for a channel capacity of up to 13.8 bits per sifted photon for high-dimensional quantum communication" and "We also propose a practical and scalable mode-multiplexed QKD protocol that cannot be achieved by alternative methods." I am therefore unable to give a final recommendation unless the authors are able to clarify my concerns below:

- In itself, I do indeed find impressive the capability to transmit so many modes across such a long fiber, and I think this is quite an advance. However I am not convinced that the two main applications mentioned by the authors can be directly carried out. Since these two applications work very differently, and for the sake of clarity, I will address each separately.

Response: We thank the reviewer for the kind words. Please see below the point-by-point responses to the questions raised by the reviewer.

Reviewer #1 — comment #2: (a) In the case of spatial high-dimensional quantum communication, and taking BB84 as an example, it is required that Alice chooses randomly between one of the two bases (HG or LG) and between any of the 210 modes in that basis. The issue is that then the time reversal system cannot transmit that specific mode from Bob to Alice, since it is not known a priori which mode it would be. And of course, if the mode to be sent was known by Bob, then there is no need to do anything since there is no information transfer. For me the problem lies in the fact that a specific mode needs to be transmitted first from Bob to Alice, and since Alice needs to send a random attenuated

spatial mode back, the time reversal compensation would not work. Therefore it is unclear to me how HD spatial QKD could work with this setup, unless if there is a single compensation that can be simultaneously applied for all modes to be transmitted. Please see my comment in the conclusion with a possible solution.

In the case of (a), the authors need to show a way to make the QC protocol working. One possibility could be to apply the compensation technique from (b) such that N different lasers would be present at Alice, and depending on which state to be sent, a different laser (corresponding to each mode) would be fired (like free-space BB84 systems where four laser diodes are used (one for each polarization state)). Maybe even using a VCSEL array perhaps, to avoid having hundreds of lasers. Or a single laser with an acousto-optic modulator? Or something else more elegant?

Response: We thank the reviewer for providing a solution to the reviewer's own question. To enable high-dimensional encoding, we agree that a scheme similar to the mode-division multiplexing can be used. As shown in the above figure, Bob needs to send N time-multiplexed spatial modes to Alice repeatedly. It should be noted that high-speed spatial mode switching as fast as 500 kHz can be readily achieved by an acousto-optic modulator to generate the time-multiplexed probe beams at Bob's side [1]. The SLM at Alice's side displays a multiplexed hologram that can simultaneously generate and multiplex a large number of beams of different incident angles as demonstrated in [2] (see the figure above). Based on the detected scrambled probe beams, Alice randomly chooses the spatial mode of her interest, fires the laser in the corresponding channel, and transmits the photon of the chosen spatial mode to Bob, which realizes the information transfer between two parties via high-dimensional encoding.

The reviewer mentions the possibility of using a VCSEL array or an acousto-optic modulator. An alternative method is to use a single laser and a 1-to- N fiber beamsplitter. For each channel after the beamsplitter, a high-speed intensity modulator can be inserted to modulate the signal. It should be noted that the loss induced by 1-to- N beam splitting is not a problem to the attenuated-coherent-state-based QKD, because strong loss is

inherently needed to attenuate the classical laser to a single-photon level. This scheme can readily be used to realize mode-division multiplexing in a similar manner.

We have included the discussions above in the revised manuscript. The changes are listed below (the revised text is in red).

(Page 4, right column, second paragraph) Figure 3(c) presents a practical, scalable QKD protocol with mode-division multiplexing. Each spatial mode can be used as an independent channel, with time-bin encoding^{55–58} or continuous-variable encoding^{59–61} used to guarantee communication security within each channel. This mode-multiplexed setup can also be used to implement high-dimensional encoding by transmitting one spatial mode at a time. The Laguerre-Gauss and Hermite-Gauss modes can be employed as mutually partially unbiased bases for high-dimensional encoding to guarantee security¹⁸.

(Page 4, right column, second paragraph) To implement the N -mode-multiplexed QKD protocol, Bob first sequentially transmits N probe beams of interest to Alice through the fiber via spatial mode switching, which can be realized by an acousto-optic modulator with a mode switching rate up to 500 kHz⁶².

(Page 5, left column, first paragraph) These N beams can be obtained by using a single laser with a 1-to- N fiber beamsplitter. The loss induced by the beamsplitter at the transmitter's side is not a concern for attenuated-coherent-state-based QKD protocols.

Reviewer #1 — comment #3: (b) The case for multiplexed-QKD is a bit more clear, but I still have some concerns. This idea in itself is not novel, and this was explored already in Bacco et al, Comm. Phys. 2, 140 (2019). There they used each spatial mode of a multi-core fiber to transmit a separate QKD stream, and then boosted the overall rate by a factor of 37. Therefore, I don't quite agree with the claim that "cannot be achieved by alternative methods", as this has already been done. Granted, the use of multi-mode fiber could boost the key even further by offering more spatial channels, but it wouldn't be the first time it was done.

Response: We are sorry that we made a misleading claim. What we intended to mean is that high-fidelity transmission of 210 spatial modes over a 1-km-long, standard, graded-index multimode fiber cannot be achieved by alternative methods (such as the slow transfer matrix inversion or the scalar phase conjugation). We are not comparing our method to multi-core fibers. We have rewritten our manuscript to avoid the possible misunderstanding. In addition, when comparing to multi-core fibers, we believe our method is still more intriguing for the following reasons.

We agree that multi-core fiber is a good platform for space-division multiplexing (here we use space-division multiplexing to specifically mean the use of multi-core fiber, and we use mode-division multiplexing to indicate the use of standard multimode fiber). However, from a technical point of view, standard multimode fibers are cheaper and already widely deployed in existing commercial fiber communication systems, and thus

the capacity of these systems can be readily improved by two orders of magnitude using our method without replacing the fibers. Furthermore, the fabrication and fusion splicing of multimode fibers are much easier than those of multi-core fibers.

From a fundamental point of view, the capability of transmitting high-fidelity spatial modes through strongly birefringent and scattering media is of interest in itself, while the use of multi-core fibers for space-division multiplexing is conceptually straightforward and does not present any new physics or methods. In our work, we generalize the scalar time reversal to the vectorial time reversal and demonstrate the effectiveness of the vectorial time reversal experimentally. The concept of time reversal is usually explained in textbooks by assuming a scalar incident field, which is an overly simplified assumption that does not work for long multimode fibers. For a scalar incident beam $\mathbf{E}_{scalar}(\mathbf{r}) = \hat{\mathbf{e}}A(\mathbf{r})e^{i\mathbf{k}\cdot\mathbf{r}}$, the scalar time reversal is expressed as (see Eq. (7.2.4) in "Nonlinear Optics", the 3rd edition)

$$\mathbf{E}_{scalar}^*(\mathbf{r}) = \hat{\mathbf{e}}^*A^*(\mathbf{r})e^{-i\mathbf{k}\cdot\mathbf{r}} \quad (1)$$

where $\hat{\mathbf{e}}$ is the polarization unit vector of the incident beam, $A(\mathbf{r})$ is the field amplitude, \mathbf{k} is the wavevector of the incident light, and $*$ denotes the complex conjugate. This equation assumes that the polarization $\hat{\mathbf{e}}$ is separable from the spatial profile $A(\mathbf{r})$. However, we clearly show in Fig. 1a in the original manuscript that the scrambled probe beam is not a separable state. Its horizontal polarization and vertical polarization have very different spatial profile. The assumption of a separable state used in textbooks is a potential reason why vectorial time reversal has received little attention from the community. In our experiment, the optical field scrambled by a long multimode fiber inevitably becomes a non-separable, vectorial field $\mathbf{E}_{vector}(\mathbf{r}) = \hat{\mathbf{x}}A_x(\mathbf{r})e^{i\mathbf{k}_x\cdot\mathbf{r}} + \hat{\mathbf{y}}A_y(\mathbf{r})e^{i\mathbf{k}_y\cdot\mathbf{r}}$, and the vectorial time reversal can be written as

$$\mathbf{E}_{vector}^*(\mathbf{r}) = \hat{\mathbf{x}}A_x^*(\mathbf{r})e^{-i\mathbf{k}_x\cdot\mathbf{r}} + \hat{\mathbf{y}}A_y^*(\mathbf{r})e^{-i\mathbf{k}_y\cdot\mathbf{r}} \quad (2)$$

where $\hat{\mathbf{x}}$ ($\hat{\mathbf{y}}$) is the horizontal (vertical) unit vector, $A_x(\mathbf{r})$ ($A_y(\mathbf{r})$) is the spatial field amplitude of the horizontally (vertically) polarized component, and \mathbf{k}_x (\mathbf{k}_y) is the wavevector of the horizontally (vertically) polarized component. Our generalization from Eq. (1) to Eq. (2) is intuitive but has important consequences. As discussed in our manuscript, the scalar time reversal has a poor mode fidelity (41.2% for Laguerre-Gaussian modes), and the vectorial time reversal has to be used to achieve a high mode fidelity (85.6% for Laguerre-Gaussian modes). Our results also confirm the time reversal symmetry of beam propagation through a 1-km-long fibers for the first time, which can be helpful to the modeling and understanding of long multimode fibers.

Finally, our method is potentially useful for other applications such as classical communication, multimode-fiber-based endoscopy [3] and lens-less microscopy [4]. We

believe that our method is more intriguing than the multi-core fibers both technically and fundamentally.

We have cited the reference mentioned by the reviewer and have included the discussions above in the revised manuscript. The changes are listed below (the revised text is in red).

(Abstract) We also propose a practical and scalable mode-multiplexed QKD protocol, which cannot be achieved by alternative methods over a standard multimode fiber.

(Page 1, right column, last paragraph) The textbook description of time reversal³³ assumes a scalar incident field $E_{scalar}(\mathbf{r}) = \hat{\mathbf{e}}A(\mathbf{r})e^{i\mathbf{k}\cdot\mathbf{r}}$, where $\hat{\mathbf{e}}$ is the polarization unit vector, $A(\mathbf{r})$ is the complex field amplitude, \mathbf{r} is the position vector, and \mathbf{k} is the wavevector of the incident field. It can be seen that the polarization $\hat{\mathbf{e}}$ and the complex field amplitude $A(\mathbf{r})$ are separable from each other. The scalar time reversal of the incident field can be expressed as³³

$$\mathbf{E}_{scalar}^*(\mathbf{r}) = \hat{\mathbf{e}}^*A^*(\mathbf{r})e^{-i\mathbf{k}\cdot\mathbf{r}}$$

where $*$ in the superscript denotes the complex conjugate. Here we generalize the scalar time reversal and propose the vectorial time reversal. For a vectorial incident field $\mathbf{E}_{vector}(\mathbf{r}) = \hat{\mathbf{x}}A_x(\mathbf{r})e^{i\mathbf{k}_x\cdot\mathbf{r}} + \hat{\mathbf{y}}A_y(\mathbf{r})e^{i\mathbf{k}_y\cdot\mathbf{r}}$, its vectorial time reversal can be written as

$$\mathbf{E}_{vector}^*(\mathbf{r}) = \hat{\mathbf{x}}A_x^*(\mathbf{r})e^{-i\mathbf{k}_x\cdot\mathbf{r}} + \hat{\mathbf{y}}A_y^*(\mathbf{r})e^{-i\mathbf{k}_y\cdot\mathbf{r}}$$

where $\hat{\mathbf{x}}$ and $\hat{\mathbf{y}}$ denote the horizontal and vertical unit vector, respectively. Here, the incident vectorial field is described by a non-separable state, which is a more general form than the separable state. All experimental demonstrations in MMFs⁴¹⁻⁴⁶ to date have been solely based upon scalar time reversal. However, the scalar time reversal can only succeed in a short MMF (≈ 1 m)⁴¹⁻⁴⁵ or a few-mode fiber⁴⁶, because the optical field scrambled by a long MMF has to be described by a vectorial field due to the spatially-varying birefringence in a MMF..

(Page 3, left column, last paragraph) It can be seen that the scrambled probe beams are vectorial fields and cannot be described by a separable state, because field profiles of horizontal and vertical polarization are very different.

(Page 5, right column, last paragraph) While specialty fibers such as multi-core fibers⁶⁶ can also be used to realize space-multiplexed QKD, standard multimode fibers are cheaper and already widely deployed in existing commercial fiber communication systems, and thus the capacity of these systems can be readily improved by two orders of magnitude using our method without replacing the fibers. Our results also confirm the time reversal symmetry of beam propagation over a 1-km-long MMF for the first time, which can be intriguing to the modeling and understanding of long MMFs. Given the scalability of the experimental implementation and high fidelity of the data, our technique can be useful to not only mode-division multiplexing but also to other applications such as fiber endoscopy⁶⁸, lensless microscopy⁶⁹, and high-dimensional entanglement distribution^{20,21}.

Reviewer #1 — comment #4: - One minor suggestion I have in the supplemental material is that the authors put the figures in line with the text where they are cited, as it makes the reading easier.

Response: We thank the reviewer for this suggestion. We have revised our supplementary material accordingly.

Reviewer #1 — comment #5: In the case of (b) I agree it looks feasible, but the impact is much more limited than 2 orders of magnitude, since it was already done gaining more than 1 order of magnitude compared to standard QKD systems (D. Bacco et al).

Response: This comment is equivalent to Reviewer#1-comment#3. Please refer to the response there.

Reviewer #1 — comment #6: In conclusion: I think the authors have a very nice result, but it is not very clear how it can be applied for other applications. The main issue for me is the requirement that each mode needs to be specifically transmitted from Bob to Alice, and that then a new beam has to be generated for that particular mode.

Response: We thank the reviewer for the positive attitude towards our work. We believe that we have addressed all concerns from the reviewer in the above responses, and we sincerely hope that the reviewer can kindly reconsider our manuscript.

Report of reviewer #2 (in blue) followed by a detailed response to each point (in black)

Reviewer #2 — summary: The paper reports an experimental study of multi-modal propagation in 1-km fiber in a context of quantum communication. The Authors describe the method named “vectorial time reversal”, which assumes that the mode transformation is first measured using off-axis holography and then the wavefront of the field is modified using spatial light modulator to reverse the effect of mode mixing. The method is demonstrated experimentally using laser light. It is also analyzed for the application in quantum key distribution. The Authors claim the capacity of the channel of the order of 13 bits per sifted photon.

Response: We thank the reviewer for the summary of our work. Please see below the point-by-point responses to the questions raised by the reviewer.

Reviewer #2 — comment #1: Here is a list of my comments:

1) The Authors present a classical method which can be applied to enhance quantum communication protocols including QKD. In my opinion this is nice piece of engineering, but I do not see new physics.

Response: The new physics is that we propose and demonstrate the effectiveness of vectorial time reversal, while the conventional method introduced in textbooks is based on scalar time reversal. Our results confirm the time reversal symmetry of beam propagation through a 1-km-long multimode fibers for the first time, which can only be achieved by vectorial time reversal. These results can be intriguing to the modeling and understanding of multimode fibers.

In textbooks that discuss the principles of time reversal, a scalar incident field is conventionally assumed, which is not a general assumption. In chapter 7 of "Nonlinear Optics" authored by Prof. Robert Boyd, the scalar time reversal is expressed as (see Eq. (7.2.4) in "Nonlinear Optics", the 3rd edition)

$$\mathbf{E}_{scalar}^*(\mathbf{r}) = \hat{\mathbf{e}}^* A^*(\mathbf{r}) e^{-i\mathbf{k}\cdot\mathbf{r}} \quad (1)$$

where $\hat{\mathbf{e}}$ is the polarization unit vector of the incident beam, $A(\mathbf{r})$ is the field amplitude, \mathbf{k} is the wavevector of the incident light, and $*$ denotes the complex conjugate. This equation assumes that the polarization $\hat{\mathbf{e}}$ is separable from the spatial profile $A(\mathbf{r})$. However, we experimentally show in Fig. 1a in the original manuscript that the scrambled probe beam is not a separable state. Its horizontal polarization and vertical polarization have very different spatial profile. The assumption of a separable state used in the textbook is a potential reason why vectorial time reversal has received little attention from the community. In our experiment, the optical field scrambled by a long multimode fiber inevitably becomes a non-separable, vectorial field $\mathbf{E}_{vector}(\mathbf{r}) = \hat{\mathbf{x}}A_x(\mathbf{r})e^{i\mathbf{k}_x\cdot\mathbf{r}} + \hat{\mathbf{y}}A_y(\mathbf{r})e^{i\mathbf{k}_y\cdot\mathbf{r}}$, and the vectorial time reversal can be written as

$$\mathbf{E}_{vector}^*(\mathbf{r}) = \hat{\mathbf{x}}A_x^*(\mathbf{r})e^{-i\mathbf{k}_x\cdot\mathbf{r}} + \hat{\mathbf{y}}A_y^*(\mathbf{r})e^{-i\mathbf{k}_y\cdot\mathbf{r}} \quad (2)$$

where $\hat{\mathbf{x}}$ ($\hat{\mathbf{y}}$) is the horizontal (vertical) unit vector, $A_x(\mathbf{r})$ ($A_y(\mathbf{r})$) is the spatial field amplitude of the horizontally (vertically) polarized component, and \mathbf{k}_x (\mathbf{k}_y) is the wavevector of the horizontally (vertically) polarized component. Our generalization from Eq. (1) to Eq. (2) is intuitive but has important consequences. As discussed in our manuscript, the scalar time reversal has a poor mode fidelity (41.2% for Laguerre-Gaussian modes), and the vectorial time reversal has to be used to achieve a high mode fidelity (85.6% for Laguerre-Gaussian modes). Our results also confirm the time reversal symmetry of beam propagation through a 1-km-long fibers for the first time, which can be intriguing to the modeling and understanding of multimode fibers. Therefore, we believe our work does present new physics.

Finally, as the reviewer said, our work is also a nice piece of engineering. Our results reveal the possibility of performing mode-multiplexed QKD through multimode fibers of an appreciable length, which cannot be achieved by alternative methods (such as the slow transfer matrix inversion, the scalar time reversal, or the classical MIMO algorithm). Therefore, we sincerely hope the reviewer can reconsider our manuscript based on the above clarifications.

We have included the discussions above in the revised manuscript. The changes are listed below (the revised text is in red).

(Page 1, right column, last paragraph) The textbook description of time reversal³³ assumes a scalar incident field $E_{scalar}(\mathbf{r}) = \hat{\mathbf{e}}A(\mathbf{r})e^{i\mathbf{k}\cdot\mathbf{r}}$, where $\hat{\mathbf{e}}$ is the polarization unit vector, $A(\mathbf{r})$ is the complex field amplitude, \mathbf{r} is the position vector, and \mathbf{k} is the wavevector of the incident field. It can be seen that the polarization $\hat{\mathbf{e}}$ and the complex field amplitude $A(\mathbf{r})$ are separable from each other. The scalar time reversal of the incident field can be expressed as³³

$$\mathbf{E}_{scalar}^*(\mathbf{r}) = \hat{\mathbf{e}}^*A^*(\mathbf{r})e^{-i\mathbf{k}\cdot\mathbf{r}}$$

where $*$ in the superscript denotes the complex conjugate. Here we generalize the scalar time reversal and propose the vectorial time reversal. For a vectorial incident field $\mathbf{E}_{vector}(\mathbf{r}) = \hat{\mathbf{x}}A_x(\mathbf{r})e^{ik_x\cdot\mathbf{r}} + \hat{\mathbf{y}}A_y(\mathbf{r})e^{ik_y\cdot\mathbf{r}}$, its vectorial time reversal can be written as

$$\mathbf{E}_{vector}^*(\mathbf{r}) = \hat{\mathbf{x}}A_x^*(\mathbf{r})e^{-ik_x\cdot\mathbf{r}} + \hat{\mathbf{y}}A_y^*(\mathbf{r})e^{-ik_y\cdot\mathbf{r}}$$

where $\hat{\mathbf{x}}$ and $\hat{\mathbf{y}}$ denote the horizontal and vertical unit vector, respectively. Here, the incident vectorial field is described by a non-separable state, which is a more general form than the separable state. All experimental demonstrations in MMFs^{41–46} to date have been solely based upon scalar time reversal. However, the scalar time reversal can only succeed in a short MMF (≈ 1 m)^{41–45} or a few-mode fiber⁴⁶, because the optical field scrambled by a long MMF has to be described by a vectorial field as we will show later.

(Page 3, left column, last paragraph) It can be seen that the scrambled probe beams are vectorial fields and cannot be described by a separable state, because field profiles of horizontal and vertical polarization are very different.

(Page 5, right column, last paragraph) Our results also confirm the time reversal symmetry of beam propagation over a 1-km-long MMF for the first time, which can be intriguing to the modeling and understanding of long MMFs. Given the scalability of the experimental implementation and high fidelity of the data, our technique can be useful to not only mode-division multiplexing but also to other applications such as fiber endoscopy⁶⁸, lensless microscopy⁶⁹, and high-dimensional entanglement distribution^{20,21}.

Reviewer #2 — comment #2: 2) In practice, photon loss is crucial for QKD applications. The method is based on adaptative optics, in particular on SLM, which are very lossy. One can argue that deformable mirror can be used, with much higher reflection and also

refresh rate, but with the limited resolution. This is an engineering problem that should be addressed properly if this method is to be proved useful in practice.

Response: We thank the reviewer for asking this question, and we are sorry that we didn't clearly answer this question in the previous manuscript. In fact, the photon loss is not a problem to our protocol. As can be seen in the figure above (Fig. 3c in the manuscript), we propose to use a densely encoded computer-generated hologram on a single spatial light modulator (SLM) to simultaneously generate and multiplex the vectorial time reversal of scrambled probe beams. Although the diffraction efficiency of densely encoded hologram is low, this is not a problem because the SLM is located at the transmitter's side. For attenuated-coherent-state-based QKD protocols, strong loss is inherently needed to attenuate a classical laser to a single-photon level. Therefore, we believe the loss is not a problem to our protocol, and the secure key rate will not be compromised by the photon loss of SLM.

We have included the discussions above in the revised manuscript. The changes are listed below (the revised text is in red).

(Page 5, left column, first paragraph) **Although the densely encoded hologram has a low efficiency, this will not reduce the secure key rate of the attenuated-coherent-state-based QKD because strong loss is inherently necessary to attenuate a classical laser to the single-photon level.**

Reviewer #2 — comment #3: 3) On page 4, just at the beginning it is said "we believe that the fidelity of time-reversed modes can be further increased by using a well-calibrated SLM". Why the proper calibration has not been done?

Response: The SLMs from different manufacturers have very different performances, and we do find that our SLM is poorly pre-calibrated by the manufacturer. We have performed preliminary calibration to our SLM by adding a few basic Zernike polynomials as the phase correction, and thus we can achieve a decent mode fidelity over a highly scattering, 1-km-long multimode fiber. To further increase the mode fidelity, one can use high-precision, high-resolution Michelson interferometry to carefully measure the residual aberration of

the SLM. Moreover, our experiment is a proof-of-principle demonstration, and we believe that a more careful calibration would not reveal additional physics or insights. We wish the reviewer can understand us and can reconsider our manuscript.

Reviewer #2 — comment #4: 4) "In this proof-of-principle experiment, we implement digital vectorial time reversal for one spatial mode at a time, but we emphasize that our method can be readily used to enable mode-division multiplexing". It would be interesting, from the perspective of practical application of the method, what is the expected performance when available adaptative optics is used (refresh rate impact mainly).

Response: We have tried to discuss the impact of the refresh rate of SLM in the original manuscript, and we are sorry that we didn't make it clear.

First we want to explain that the refresh rate of SLM does not limit the data transfer rate. As explained in the abstract in [5], "*The beam encoding/decoding speed is not limited by the conventional slow switching response of a spatial light modulator (SLM) but is fully determined by the modulation rate of an intensity modulator, which easily supports tens of gigabits per second modulation and resultant encoding/decoding*".

Second, we emphasize that the refresh rate of SLM only needs to be faster than the environmental fluctuation rate in order to overcome instability. In our experiment, we monitor the fidelity fluctuation through the fiber for 20 minutes, and the correlation time is on the order of tens of seconds. Because multimode fibers are well protected in commercial systems and are typically not susceptible to strong vibrations, the commercially available digital micromirror device (above 10 kHz refresh rate) as well as the fast SLM (sub-kHz refresh rate) will be sufficient to stabilize the fiber link.

Therefore, we believe that commercially available equipment can readily be used to implement our protocol in realistic conditions. We have included the discussions above in the revised manuscript. The changes are listed below (the revised text is in red).

(Page 4, right column, first column) **In addition, we emphasize that the data transfer rate using each spatial mode is not limited by the refresh rate of SLM or the response time of the time-reversal system but is determined by the modulation rate of the modulator used by the encoder⁵⁴, and the time-reversal system response time only needs to be faster than the environmental fluctuation rate in order to overcome instability.**

Reviewer #2 — comment #5: 5) There is multiple trade-offs, which are not discussed, including: the photon loss decreases the efficiency of QKD but the channel capacity is increased by using higher spatial modes.

Response: This comment is equivalent to Reviewer#2-comment#2. Please refer to our response there. The conclusion is that the photon loss of the SLM does not decrease the

efficiency of QKD, because strong loss is inherently needed to attenuate a classical laser to a single-photon level for a attenuated-coherent-state-based QKD protocol.

Reviewer #2 — comment #6: 6) In the paper [Optics Express, 2019, 27, 37214] the authors present a passive method of reducing an effect of multi-modal propagation. It is related to free space propagation, but the underlying physics is related.

Response: We thank the reviewer for bringing this reference to our attention. This work uses time-bin encoding, and the spatial degree of freedom is not used for multiplexing in their protocol. These authors use multimode fiber to increase the collection efficiency of the photons whose spatial profile is scrambled by turbulence, and no active adaptive optics is used to correct the turbulence-induced aberration. However, we do find that this reference can be helpful since time-bin encoding is also applicable to our method. Therefore, we have included this reference to help the readers who want to implement time-bin QKD.

We have included the discussions above in the revised manuscript. The changes are listed below (the revised text is in red).

(Page 4, right column, last paragraph) Each spatial mode can be used as an independent channel, with time-bin encoding⁵⁵⁻⁵⁸ or continuous-variable encoding⁵⁹⁻⁶¹ used to guarantee communication security within each channel.

Reviewer #2 — comment #7: Minor comments:

1) Page 1, column 2 "However, it require a high signal-to-noise ratio..." should be "requires"

2) page 2, column 1 "Figure 1(b) presents the the conceptual schematic.." one "the" should be dropped.

Response: We thank the reviewer for pointing out these mistakes, and we have implemented them in the revised manuscript.

Reviewer #2 — comment #8: Summarizing, the paper and supplementary material are very clearly written. The presented method does not offer any significant improvement for quantum communication protocols (eg for QKD). Some further analysis is required. I do not recommend this manuscript to publish in Nature Communication. I suggest much more specialized journal.

Response: We thank the reviewer for the positive words to our writing. The reviewer claims that the photon loss compromises the capacity enhancement and thus our method does not offer significant improvement for QKD. We have clarified that the loss is not a problem in the responses for Reviewer#2-comment#2. Given these clarifications, we

believe that our method can offer significant improvement for QKD, and we sincerely hope that the reviewer can re-evaluate our manuscript.

Report of reviewer #3 (in blue) followed by a detailed response to each point (in black)

Reviewer #3 — comment #1: The paper “High-fidelity spatial mode transmission through a 1-km-long multimode fiber via vectorial time reversal”, presents an experimental demonstration of a rudimental scheme for quantum key distribution (QKD).

The authors would like to exploit the space encoding technique for transmitting high-dimensional quantum states over 1 km of multimode fiber combined with space-division multiplexing.

Although the work looks interesting for the entire quantum community and the experiment is documented with rigour, this paper is not demonstrating any advantage compared to standard QKD scheme. The secret key is not measured, neither extrapolates since all the measurements are done in the classical domain and the scalability of the entire scheme is not feasible.

Response: We thank the reviewer for the summary of our work. However, we respectfully disagree with the statement (1) that “*this paper is not demonstrating any advantage compared to standard QKD scheme*” and (2) that “*the scalability of the entire scheme is not feasible*”. We answer these comments point-by-point as detailed below.

(1) Standard multimode fibers have been widely used in existing commercial communication systems, and it has been known for decades that the large number of spatial modes naturally provide a platform to significantly boost the speed of QKD by mode-division multiplexing. However, the modal crosstalk remains the most difficult and long-standing challenge that persistently prohibits the mode-multiplexed QKD over standard multimode fibers. Indeed, our manuscript does not propose any new type of QKD protocol, and we just follow the well-known protocols (e.g., the decoy-state QKD). However, the vectorial time reversal presented in our manuscript immediately allows for boosting the speed of QKD by a factor of approximately 210 via mode-division multiplexing. Indeed, when using attenuated lasers and single-photon detectors, the crosstalk will be slightly higher because of the detector noise. But the detector noise is totally irrelevant to vectorial time reversal. The high spatial fidelity presented in our manuscript is the most important advantage that can only be obtained by our method over a 1-km-long standard multimode fiber. For this reason, we respectfully disagree with the statement that “*this paper is not demonstrating any advantage compared to standard*

"QKD scheme", and we believe that the classical measurements performed in our experiments do not reduce the novelty of our work.

(2) We are not sure why the reviewer thinks that "the scalability of the entire scheme is not feasible", because no explicit reason is given by the reviewer. In contrast, we believe our scheme is scalable, and the proposed schematic is shown above (Fig. 3c in the original manuscript). In this schematic, Alice displays a densely encoded computer-generated hologram on the SLM to simultaneously generate and multiplex the generated time-reversed modes. It should be noted that such multiplexed hologram has been experimentally demonstrated for 105 spatial modes [2]. Although the densely encoded hologram has a low diffraction efficiency, this is not a problem to attenuated-coherent-state-based QKD protocols, because strong loss is inherently needed to attenuate a classical laser to a single-photon level. Based on these facts, we do think that our scheme is scalable, and it is straightforward to implement our protocol. We sincerely hope that the reviewer can re-consider our manuscript based on these clarifications.

Reviewer #3 — comment #2: My main concern is that the authors claim to do QKD without using single photon detectors at all. QKD does not only work by attenuating a laser and doing classical measurements.

One of the main difficulties of high-dimensional quantum key distribution is to prepare quantum states in real-time. All the claims about high-rate and good fidelity are not sufficient since when you are preparing your quantum states in real-time you will have different errors and different performances. Moreover, the authors proposed a scheme for multiplexing which in principles should work, but then they don't have a single QKD system to be tested on their experiment.

As such I recommend the authors to carefully consider the journals in which they would like to publish their work.

Response: We agree with the reviewer that we didn't use any single-photon detectors and that we didn't build a complete QKD system with real-time quantum state generation. As a response to this comment, we have removed our claim "a channel capacity of up to 13.8 bits per sifted photon" to avoid possible misunderstanding.

However, we respectfully disagree that this should be the major concern for rejecting our manuscript. We wish to emphasize that the modal crosstalk is the most outstanding challenge that prohibits the implementation of mode-multiplexed QKD over standard multimode fibers. This long-standing challenge can now be overcome exclusively by our vectorial time reversal technique, which is the most important novelty of our manuscript. By contrast, the construction of a complete QKD system with single-photon detectors has been demonstrated by many previous publications, and using single-photon detectors will not help reveal new physics in our experiments. The detector noise is fundamentally irrelevant to the modal crosstalk measured in our experiment. In addition, our method is useful for not only QKD but also other classical applications such as classical communication, multimode-fiber-based endoscopy [3] and lens-less microscopy [4]. Therefore, we believe that the classical measurements performed in our experiments do not reduce the novelty and significance of our work.

The reviewer also raises a concern about the errors induced by real-time quantum state generation. In our experiment, we already present the data when the SLM is in real-time operation as shown in the figure above, and the mode fidelity remains high over 20 minutes. It should be noted that the refresh rate of SLM only needs to be faster than the environmental fluctuation rate in order to overcome instability. In our experiment, we monitor the fidelity fluctuation through the fiber for 20 minutes, and the correlation time is on the order of tens of seconds. Therefore, the hologram on the SLM only needs to be updated at a speed of a few hertz, which can be done by commercially available SLMs. On the other hand, high-speed modulation is performed by the "QKD encoder" as shown in the figure below. Commercial intensity modulators can be used to realize the QKD encoder, and the modulation speed can be well above GHz. Therefore, the hologram on the SLM is almost static to the QKD encoder. Moreover, while we agree that additional errors might occur when preparing quantum states in real time, the errors induced by high-speed state generation are fundamentally irrelevant to the vectorial time reversal

technique demonstrated in our manuscript. Therefore, the novelty of our work should not be reduced for this reason.

In brief, the suppression of modal crosstalk in a 1-km-long standard multimode fiber by using vectorial time reversal is the most significant novelty in our manuscript, and this achievement cannot be realized by other existing methods over standard multimode fibers. Based on our result, the implementation of a complete QKD system is merely an engineering work and can be done by following previous publications. Hence, we sincerely hope that the reviewer can reconsider the value of our manuscript.

We have included the discussions above in the revised manuscript. The changes are listed below (the revised text is in red).

(Abstract) Through the use of vectorial time reversal, we show an average mode fidelity above 80% for a fiber without thermal or mechanical stabilization, ~~allowing for a channel capacity of up to 13.8 bits per sifted photon for high-dimensional quantum communication.~~

(Page 3, right column, first paragraph) ~~Although the performance is characterized using a classical light source and a classical detector, our method is readily applicable to QKD by attenuating the light intensity to a single-photon level and by using single-photon detectors⁵⁰. The additional noise of single-photon detectors is outside the scope of this work. Based on the measured crosstalk matrices, we can achieve a channel capacity of 13.8 bits per sifted photon with Laguerre-Gauss modes and 13.4 bits per sifted photon with Hermite Gauss modes for high-dimensional QKD with spatial mode encoding (see Supplementary Section 3).~~

Reviewer #3 — comment #3: Here I attached some more detailed comments:

1. The quantum states which are used are not clearly reported in the manuscript.

Response: In our manuscript, we proposed a setup for mode-multiplexed QKD in Fig. 3c, and the spatial modes are used as independent channels for multiplexing and demultiplexing. For mode-division multiplexing, the quantum states that are used for

secure information transfer depend on the specific encoding protocols to be used, such as the time-bin encoding or the continuous-variable encoding.

The same setup can also be used for high-dimensional spatial mode encoding, and the quantum states in this case are the well-known Laguerre-Gauss modes and Hermite-Gauss modes. It should be noted that the Laguerre-Gauss and Hermite-Gauss modes can be used as partially mutually unbiased bases to guarantee security [6].

We thank the reviewer for pointing this out and we have included the discussions above in the revised manuscript. The changes are listed below (the revised text is in red).

(Page 4, right column, last paragraph) Each spatial mode can be used as an independent channel, with time-bin encoding⁵⁵⁻⁵⁸ or continuous-variable encoding⁵⁹⁻⁶¹ used to guarantee communication security within each channel. This mode-multiplexed setup can also be used to implement high-dimensional encoding by transmitting one spatial mode at a time. The Laguerre-Gauss and Hermite-Gauss modes can be employed as mutually partially unbiased bases for high-dimensional encoding to guarantee security¹⁸.

Reviewer #3 — comment #4: 2. Decoy state technique is not included at all in your analysis. Why not?

Response: In our original manuscript, we did mention that several different QKD protocols (such as decoy state technique, continuous-variable QKD, etc) can be used in our scheme to guarantee communication security. However, in our response to Reviewer#3-comment#3, we have emphasized that the core achievement of our manuscript is the excellent modal crosstalk suppression realized by our vectorial time reversal method. Based on this method, it is straightforward to implement well-known protocols such as decoy-state QKD, time-bin QKD, etc. Therefore, the analysis of the decoy state is irrelevant to the vectorial time reversal and the spatial mode fidelity. Hence, we didn't included it in our manuscript.

Reviewer #3 — comment #5: 3. A stability measurement over time is missing in the paper. How is drifting the system?

Response: We are not sure why the reviewer says that “a stability measurement over time is missing”. We did present the stability measurement in Fig. 3b in the original manuscript (see the figure above). It can be clearly seen that the fidelity remains high in the presence of vectorial time reversal over 20 minutes, while the modal crosstalk immediately increases in the absence of vectorial time reversal. These results clearly show that our system is able to overcome environmental instability even though the unprotected 1-km-long bare fiber is placed on an optical table that has not been floated and is free of any thermal or mechanical isolation. The image of the unprotected fiber spool is shown in Supplementary Fig. S1.

We have included the discussions above in the revised manuscript. The changes are listed below (the revised text is in red).

(Page 4, right column, first paragraph) **These results clearly show that our system is able to overcome environmental instability even though the unprotected 1-km-long bare fiber is placed on an optical table that has not been floated and is free of any thermal or mechanical isolation. The image of the unprotected fiber spool is shown in Supplementary Fig. S1.**

Reviewer #3 — comment #6: 4. In the quantum systems the losses are very important parameters? How many losses do you have at your receiver?

Response: We agree that losses are important to QKD. Here we show the proposed QKD protocol in the above figure (Fig. 3c in the original manuscript). Compared to other standard QKD systems, we need a beamsplitter (BS) at the receiver’s side. Therefore, the loss induced by this BS will decrease the secure key rate. However, we emphasize that the loss of this BS can be mitigated by using a high-transmittance BS, such as a T=95%, R=5% BS. In this case, our system will have 5% higher loss than other QKD systems. This additional loss can be well compensated by the capacity improvement achieved by mode-division multiplexing. For other components at the receiver (such as the mode sorter and QKD decoder), the loss solely depends on the specifications of the equipment and can always be reduced by using high-quality optics with appropriate anti-reflection coatings. We thank the reviewer for pointing out the loss problem and we have included this discussion in the revised manuscript.

We have included the discussions above in the revised manuscript. The changes are listed below (the revised text is in red).

The loss induced by the beamsplitter at the receiver's side can be reduced by using a high-transmission beamsplitter (such as a T:R=95:5 beamsplitter). We note that the reduced secure key rate caused by this small amount of additional loss can be well compromised by the capacity improvement of mode-division multiplexing.

Reviewer #3 — comment #7: 5. Since the authors are claiming to use this setup for QKD, how would you modulate these optical modes? With a spatial-light-modulator at a very low-speed? Do you have other solutions?

Response: We thank the reviewer for bringing up this question. It is a common misunderstanding that the communication rate is limited by the slow refresh rate of SLM. Here we quote the abstract of [5]: *"The beam encoding/decoding speed is not limited by the conventional slow switching response of a spatial light modulator (SLM) but is fully determined by the modulation rate of an intensity modulator, which easily supports tens of gigabits per second modulation and resultant encoding/decoding"*. In the above figure, the modulator is inside the "High-speed QKD encoder", and the refresh rate of SLM only needs to be faster than the environmental fluctuation rate in order to overcome instability. We have included these discussions in the revised manuscript.

We have included the discussions above in the revised manuscript. The changes are listed below (the revised text is in red).

In addition, we emphasize that the data transfer rate using each spatial mode is not limited by the refresh rate of SLM or the response time of the time-reversal system but is determined by the modulation rate of the modulator used by the encoder⁵⁴, and the time-reversal system response time only needs to be faster than the environmental fluctuation rate in order to overcome instability.

We believe that our responses have addressed the reviewer's concerns and we sincerely hope that the reviewer can reconsider our manuscript.

References

- [1] Braverman, B., Skerjanc, A., Sullivan, N., & Boyd, R. W. Fast generation and detection of spatial modes of light using an acousto-optic modulator. *Optics Express*, 28(20), 29112-29121. (2020).
- [2] Trichili, A. et al. Optical communication beyond orbital angular momentum. *Scientific Reports* 6, 27674 (2016).
- [3] Turtaev, S. et al. High-fidelity multimode fibre-based endoscopy for deep brain in vivo imaging. *Light Sci. Appl.* 7, 1–8 (2018).
- [4] Čižmár T, Dholakia K. Exploiting multimode waveguides for pure fibre-based imaging. *Nature Communications*, 3(1): 1-9. (2012).
- [5] Chen, S., et al. Demonstration of 20-Gbit/s high-speed Bessel beam encoding/decoding link with adaptive turbulence compensation. *Optics Letters*, 41(20), 4680-4683. (2016).
- [6] Wang, F. et al. High-dimensional quantum key distribution based on mutually partially unbiased bases. *Physical Review A*, 101(3), 032340. (2020).

Reviewers' Comments:

Reviewer #1:

Remarks to the Author:

I went over the resubmitted manuscript and I am happy with most of the modifications. In my opinion, the main achievement of this paper is the transmission of a large number of spatial modes over a multi-mode optical fibre of considerable length. The main change now is that the authors have clarified how their technique may be applied to practical scenarios in quantum communication. This was an important step to justify publication.

As before my view of the result is that it is quite broad and relevant in other areas as well, such as imaging, so these results could definitely be published in a multi-disciplinary journal with a large audience such as Nature Communications. However, before giving my final recommendation there is one remark the authors should include in their conclusions to give it a fairer view to the readers:

Although the fact that 420 modes, divided between two mutually unbiased bases, could be transmitted, the fidelity of any particular mode taken at random is not very high. Therefore, it is unlikely that long distances could be achieved due to the error rates that would likely only increase when all the modes need to be encoded in parallel. This study is beyond the scope of this paper, I agree. In any case, the authors should include, in my opinion, a statement that explains that further studies are probably needed to expand the transmission distance further since 1 km limits the applications only to local area networks.

Reviewer #2:

Remarks to the Author:

I share opinion of the other Referees that the experiment is technically sound and impressive. However, the claim that the vectorial time reversal as an "intuitive", as the Authors said, generalization of scalar time reversal is new physics is over hyped. I do agree that the method indeed required vectorial approach as scalar one, obviously, would not work because of the correlation.

From technological point of view, 1 km of fiber is not a real life distance. Although, the results indicate that the method has a strong potential to be applied in realistic scenarios.

The response and the updated version of the manuscript does not add anything new, which would result in a change of my opinion. I suggest again the Authors should consider more specialized journal for their nice work.

Reviewer #3:

Remarks to the Author:

The new version of the paper is well written and includes most of my comments. However, after the revision, I am still convinced that the claim is too big compared to the experiment reported in this paper. I do agree that the paper shows an interesting method to decrease the modal cross-talk and this method could be useful in quantum communication, classical communication, etc.. but the authors did not demonstrate any of these applications. The experiment is a characterization of the optical fibre, and although the method is interesting and could be used in multiple applications, the absence of one specific demonstration is lowering the overall impact of the paper.

I am confident to recommend the publication in a more technical journal.

However, I would be happy to reconsider my opinion if the authors, in the future, will eventually demonstrate a multiplexing or high-dimensional QKD protocol using their new method.

Response Letter #2 (NCOMMS-20-31935A-Z)

Report of reviewer #1 (in blue) followed by a detailed response to each point (in black)

Reviewer #1 — comment #1: I went over the resubmitted manuscript and I am happy with most of the modifications. In my opinion, the main achievement of this paper is the transmission of a large number of spatial modes over a multi-mode optical fibre of considerable length. The main change now is that the authors have clarified how their technique may be applied to practical scenarios in quantum communication. This was an important step to justify publication.

As before my view of the result is that it is quite broad and relevant in other areas as well, such as imaging, so these results could definitely be published in a multi-disciplinary journal with a large audience such as Nature Communications. However, before giving my final recommendation there is one remark the authors should include in their conclusions to give it a fairer view to the readers:

Response: We thank the reviewer for the positive words. Please see below the point-by-point responses to the questions raised by the reviewer.

Reviewer #1 — comment #2: Although the fact that 420 modes, divided between two mutually unbiased bases, could be transmitted, the fidelity of any particular mode taken at random is not very high. Therefore, it is unlikely that long distances could be achieved due to the error rates that would likely only increase when all the modes need to be encoded in parallel. This study is beyond the scope of this paper, I agree. In any case, the authors should include, in my opinion, a statement that explains that further studies are probably needed to expand the transmission distance further since 1 km limits the applications only to local area networks.

Response: We agree that 1 km limits the applications only to local area networks, and we have made the statement that further studies are probably needed to expand the transmission distance further. However, we believe that our result is still a breakthrough for the following reasons:

(1). In the table below, we summarize recent works on mode transmission over standard multimode fibers. It can be seen that our 1-km fiber length is nearly three orders of magnitude larger than fiber lengths used in all other experiments demonstrating high-fidelity mode transmission through multimode fibers. Although mode-group excitation has been demonstrated over long fibers, this method only allows for the transmission of a few mode groups (less than 10) and thus is not directly comparable to our method.

Based on this literature review, we believe that our vectorial time reversal method allows for state-of-the-art propagation distances over standard multimode fibers, which cannot be achieved by

existing methods to the best of our knowledge. Therefore, we believe that our work presents an important step towards practical applications over an appreciable distance.

No.	Method	Multimode fiber length	Number of demonstrated modes	Reference
1	Vectorial time reversal	1 km	210 HG/LG modes	This work
2	Transfer matrix inversion	2 m	6 discrete spot modes	Nat. Phys. 16, 1112–1116 (2020)
3	Transfer matrix inversion	5 m	N/A	Nat. Commun. 11, 5813 (2020)
4	Transfer matrix inversion	2 m	110 LP modes	Opt. Express 22, 96–101 (2014)
5	Transfer matrix inversion	0.3 m	110 LP modes	Nat. Photon. 9, 529 (2015)
6	Transfer matrix inversion	0.5 m	18 discrete spot modes	Nat. Photon. 14, 139–142 (2020)
7	Transfer matrix inversion	2 m	N/A	Phys. Rev. X 9, 041050 (2019)
8	Transfer matrix inversion	1 m	N/A	Nat. Commun. 10, 5085 (2019)
9	Transfer matrix inversion	2 m	N/A	Light Sci. Appl. 7, 54 (2018)
10	Transfer matrix inversion	1 m	N/A	Nat. Commun. 10, 2973 (2019)
11	Scalar time reversal	1 m	N/A	Opt. Express 20, 10583–10590 (2012)
12	Scalar time reversal	0.19 m	N/A	Appl. Opt. 59, 701–705 (2020)
13	Scalar time reversal	0.3 m	N/A	Opt. Express 23, 9109–9120 (2015)
14	Scalar time reversal	2 m	N/A	Opt. Express 24, 15128–15136 (2016)
15	Mode-group excitation	44.3 km	6 mode groups	Opt. Express 23, 235–246 (2015)
16	Mode-group excitation	5 km	8 mode groups	IEEE Photon. Technol. Lett. 24, 1363–1365 (2012)
17	Mode-group excitation	2.6 km	4 mode groups	Opt. Express 25, 25637–25645 (2017)

Supplementary Table 1. Summary of methods for modal crosstalk suppression in standard multimode fibers. It can be seen that transfer matrix inversion has only been applied to short fibers that are less than 5-m long. This is because the characterization of a high-dimensional transfer matrix can take as long as hours. Since it is technically challenging to stabilize long fibers for hours, short fibers are used for experimental demonstrations. It can also be seen that scalar time reversal has only been used for short fibers. In Supplementary Note 6 we show that scalar time reversal cannot be used for a 1-km-long fiber due to the inevitable polarization mixing in long fibers. Mode-group excitation can be used to transmit mode groups over a long fiber. However, a standard MMF only supports a few mode groups, and thus this method is not comparable to our approach. HG: Hermite-Gauss. LG: Laguerre-Gauss. LP: linearly polarized.

(2). As shown in Fig. 2c in the manuscript (see below), the measured mode fidelity matches well with the experimental generation fidelity for most LG modes. This suggests that the measured fidelity of these modes are limited by our experimental apparatus (e.g., the spatial light modulator) rather than the fiber length. Therefore, we expect that similar fidelity could be obtained when using a longer fiber, which is subject to a future study.

(3). It should be noted that we intentionally splice two 500-m fibers to form a 1-km-long fiber, and we can still achieve a high mode fidelity for most modes. In the above figure, we do observe that the fidelity drops for high-order modes, which is potentially caused by the fiber splicing or bending. In practical applications, we can drop the high-order modes and still achieve high fidelity for an appreciable number of modes. We agree that further studies are needed to identify the exact reason in order to further expand the transmission distance for high-order modes.

In conclusion, we believe that the 1-km fiber length is an appreciable distance, and our method has the potential to be applicable to longer fibers. In the meantime, we agree that further studies are needed to test the performance of our method over longer fibers. The revisions we have made are listed below.

Revised text at Supplementary Note 1: We have added the literature review (shown in the above table) in Supplementary Note 1. Here we skip the lengthy text for conciseness.

Revised text at page 2, right column, second paragraph: The fiber length used in our experiment is nearly three orders of magnitude greater than those used in many previously reported demonstrations (see Supplementary Note 1).

Revised text at page 3, right column, first paragraph: It can be seen that the fidelity of unscrambled signal beams received by Bob matches well with the experimental generation fidelity. Hence, the fidelity of unscrambled signal beams is limited by our experimental apparatus (e.g., the SLM) rather than the fiber, and we believe that our method can be applicable to even longer fibers.

Revised text at page 3, right column, first paragraph: Nonetheless, there exists a deviation between the unscrambled mode fidelity and the experimental generation fidelity for high-order modes, which we attribute to the fact that high-order modes are susceptible to mode-dependent loss induced by fiber bending and splicing. Further studies are needed to identify the exact reason in order to further expand the transmission distance for high-order modes.

Report of reviewer #2 (in blue) followed by a detailed response to each point (in black)

Reviewer #2 — Comment #1: I share opinion of the other Referees that the experiment is technically sound and impressive. However, the claim that the vectorial time reversal as an "intuitive", as the Authors said, generalization of scalar time reversal is new physics is over hyped. I do agree that the method indeed required vectorial approach as scalar one, obviously, would not work because of the correlation.

From technological point of view, 1 km of fiber is not a real life distance. Although, the results indicate that the method has a strong potential to be applied in realistic scenarios.

The response and the updated version of the manuscript does not add anything new, which would result in a change of my opinion. I suggest again the Authors should consider more specialized journal for their nice work.

Response: We thank the reviewer for the summary of our work. We have revised our manuscript to avoid over-hyped statements. The concern on the fiber length is similar to Reviewer #1—Comment #2. We wish to emphasize that our 1-km fiber length is nearly three orders of magnitude larger than fiber lengths used in related works demonstrating high-fidelity mode transmission through multimode fibers, and that our results should be applicable to longer fibers. For a full response, we refer the reviewer to our response to Reviewer #1—Comment #2.

Revised text at page 3, right column, first paragraph: Our results also confirm the time reversal symmetry of beam propagation over a 1-km-long MMF, ~~which can be intriguing to the modeling and understanding of long MMFs.~~

Report of reviewer #3 (in blue) followed by a detailed response to each point (in black)

Reviewer #3 — comment #1: The new version of the paper is well written and includes most of my comments. However, after the revision, I am still convinced that the claim is too big compared to the experiment reported in this paper. I do agree that the paper shows an interesting method to decrease the modal cross-talk and this method could be useful in quantum communication, classical communication, etc.. but the authors did not demonstrate any of these applications. The experiment is a characterization of the optical fibre, and although the method is interesting and could be used in multiple applications, the absence of one specific demonstration is lowering the overall impact of the paper.

I am confident to recommend the publication in a more technical journal.

However, I would be happy to reconsider my opinion if the authors, in the future, will eventually demonstrate a multiplexing or high-dimensional QKD protocol using their new method.

Response: We understand that the reviewer was expecting a complete implementation of a communication system. However, it is our opinion that implementing a complete communication system was quite reasonably not the point of our paper. Instead, we isolated the most challenging problem in implementing spatial mode multiplexing over a long multimode fiber and demonstrated how the problem could be addressed. Since our method is the only one that is applicable to long fibers to the best of our knowledge, we believe that it will impact the entire multimode fiber community and become an indispensable tool for applications based on long fibers.

We have summarized recent works on spatial mode transmission over standard multimode fibers. We wish to emphasize that our 1-km fiber length is nearly three orders of magnitude larger than fiber lengths used in all other experiments demonstrating high-fidelity mode transmission through multimode fibers. Here, we refer the reviewer to our response to Reviewer #1—Comment #2 for the details of a literature review. We believe that our method is an indispensable tool to enable previously proposed applications over long distances and thus can have a high impact.